# Cosmic Evolution of the Logarithmic $f(R)$ Model and the dS Swampland Conjecture

Jafar Sadeghi [1,2], Behnam Pourhassan [3,4,*], Saeed Noori Gashti [1], Elaheh Naghd Mezerji [1] and Antonio Pasqua [5]

1   Department of Physics, Faculty of Basic Sciences, University of Mazandaran, Babolsar P.O. Box 47416-95447, Iran
2   Canadian Quantum Research Center, 204-3002 32 Ave, Vernon, BC V1T 2L7, Canada
3   School of Physics, Damghan University, Damghan P.O. Box 3671645661, Iran
4   Physics Department, Istanbul Technical University, Istanbul 34469, Turkey
5   Department of Physics, University of Trieste, Via Valerio, 2, 34127 Trieste, Italy
*   Correspondence: b.pourhassan@du.ac.ir

**Abstract:** In this paper, we study the inflationary scenario in logarithmic $f(R)$ gravity, where the rate of inflation roll is constant. On the other hand, our gravitational $f(R)$ model is a polynomial plus a logarithmic term. We take advantage of constant-roll conditions and investigate the cosmic evolution of the logarithmic $f(R)$ gravity. We present a numerical and a graphical study using the model parameters. Additionally, we obtain the corresponding potential by using the constant-roll condition. We obtain the exact value of the potential satisfying the constant-roll conditions. Next, we challenge it with refined swampland conjecture with respect to the Planck data. Finally, we compare our results with the latest observable data.

**Keywords:** inflation; $f(R)$ gravity; cosmology; swampland conjecture

## 1. Introduction

Recently, inflation has been considered as an important concept to describe the early cosmology. Experiments in the cosmic microwave field have established several constraints on the physics of inflation. However, despite the enormous progress, a convincing and theoretically well-motivated model of inflation still lacks, in spite of the vast array of current inflation models. Furthermore, in some cases, observables are insufficient to distinguish between different models. By defining new inflation models, it was found that inflation can be implemented by a (scalar) field that rolls down on its potential under certain conditions. It is important to emphasize that the exponential expansion of the scale factor leads to an exponential temperature decreasing. Furthermore, an exponentially growing scale factor also leads to an exponentially decreasing energy density, so that at the end of inflation, the universe is extremely cold and filled only by inflation. For this reason, right after inflation ends, we need to have a phase transition where the inflation decays into other species (matter and radiation) that reassemble the universe.

Inflationary models have been proposed to address and solve several problems, such as the horizon, flatness, and the absence of magnetic monopoles [1–6]. The inflationary paradigm in cosmology is one of the plausible scenarios that describe the universe's early evolution; indeed, different inflationary models have been studied in various conditions, such as slow-roll, constant-roll, and many other structures in the literature. Several theories can describe inflation of the universe, such as modified $f(R)$ gravitational models [1,2]. Modified gravity appears in different forms and generally plays a vital role in describing the universe's evolution [4–7]. In particular, many phenomena related to different stages of evolution associated with the present universe can be investigated using modified gravity theories [8]. Recent studies in cosmology and other sciences, such as particle

physics, have led to remarkable developments. Cosmic microwave background (CMB) measurements show that fluctuations in matter and energy are always unstable on a large scale. The leading cause of these fluctuations is still unknown. Researchers have provided explanations for these fluctuations, as for all cosmological questions and problems, such as the inflationary universe. Thus, the cosmic inflation has been proposed as a great pattern for these problems. To solve these problems, the universe has gone through an early phase of accelerated expansion. It also plays a fundamental role in explaining the origin of anisotropy in the cosmic microwave background radiation and large-scale structures. Inflation models are placed in different forms and categories, and according to specific characteristics, each inflation model is classified into a particular class. The famous classification of inflation in two groups are standard inflation (cold inflation) and warm inflation. There are also several independent methods for classifying inflation models. Another type of inflationary model category is based on the initial conditions of inflation and different areas possible during the inflation, such as quasi-potential inflation, power law inflation, etc. Another possible classification is based on the end of the inflation. Various theories related to modified gravity, such as $f(R)$, lead us to find the unifying description of some periods associated with the universe as an early-time and late-time acceleration era [9–11]. The inflationary scenario has been studied from different modified gravity structures such as $f(R)$, $f(R, T)$, and $f(G)$, also with various conditions such as constant-roll, slow-roll, and ultra-slow roll [12–45]. Recently, a new idea which is called the swampland program was introduced. The swampland program includes several conjectures, such as the weak gravity conjecture (WGC), swampland dS conjecture, swampland distance conjecture, trans-Planckian censorship conjecture (TCC), etc. In the last decade, swampland conjectures have been studied in various concepts such as inflation, the physics of black holes, dark energy, etc. In physics, swampland refers to the effective low-energy physical theories incompatible with string theory, in contrast to the so-called "string theory landscape" of compatible ideas. In other words, the swampland is the set of consistent-looking theories with no consistent ultraviolet completion in string theory. Developments in string theory suggest that the string theory landscape of false vacua is vast, so it is natural to ask if the landscape is as vast as allowed by consistent-looking effective field theories. Some authors suggest that that is not the case and that the swampland is, in fact, much larger than the string theory landscape [46–64].

This paper will investigate the constant-roll evolution of modified $f(R)$ gravity, a polynomial plus a logarithmic term. Here, we aim to analyze and evaluate the scalar index spectrum $n_s$, and tensor-to-scalar ratio concerning $n$ and $\beta$ of this $f(R)$ gravity. We briefly explain and then examine our inflationary model. Therefore, in Section 2, we will first introduce the concepts and relations related to the gravitational model's evolution. In Section 3, we present the modified gravitational model and examine some corresponding relations discussed in the previous section. In this section, we also analyze the above model with some different figures. In Section 4, we investigate the potential of our logarithmic inflation model by applying the constant roll condition, and we challenge it with refined swampland conjecture. Finally, in the last section, we will explain the paper's results.

## 2. $f(R)$ Gravity and Constant-Roll Evolution

In this section, we assume that a constant-roll era occurred during the period of inflation. The inflationary paradigm of constant-roll has been used in the content of scalar-tensor theories [36–43] as well as in the generalized range of $f(R)$ modified gravity [65–67] and many have examined it in previous works. We first briefly explain $f(R)$ gravity, and then we study our inflationary model [30,33,68–72]. We assume $c = \hbar = M_{pl} = 1$, so we consider the action which is given by

$$S_J = \int d^4 X \sqrt{-g} \frac{f(R)}{2}. \tag{1}$$

The Friedmann–Lemaître–Robertson–Walker metric is as follows:

$$ds_J^2 = -dt^2 + \alpha^2(t)(dx^2 + dy^2 + dz^2), \tag{2}$$

where the above relations were in the Jordan frame, but we can transfer these relations to the Einstein framework by a conformal transformation such as $g_{\mu\nu}^E = F g_{\mu\nu}^J$. Therefore, using these conformal transformations, the above action, which is given by,

$$S_E = \int d^4 X \sqrt{-g} \left( -\frac{1}{2} R + \frac{1}{2} g^{\mu\nu} \partial_\mu \phi \partial_\nu \phi - V(\phi) \right), \tag{3}$$

where $J$ is the symbol of the Jordan frame, and subscript $E$ denotes the Einstein frame. Additionally, we will have

$$V(\phi) = \frac{1}{2} \frac{RF - f}{F^2}, \tag{4}$$

and

$$F = \frac{df}{dR}. \tag{5}$$

We want to describe the modified $f(R)$ gravity, which has an important role in describing dark energy and cosmic acceleration. The most natural extension constant-roll condition considered in most works is usually in the following form,

$$\frac{\ddot{H}}{2H\dot{H}} \simeq \beta, \tag{6}$$

where $\beta$ is a constant parameter that can have positive or negative values. Equation (6) can be written by,

$$\frac{\ddot{\phi}}{H\dot{\phi}} = \beta. \tag{7}$$

Moreover, the second slow-roll condition is $\eta \sim -\frac{\ddot{H}}{2H\dot{H}}$. We assume a theory described by $f(R)$ gravity and that the background is a flat FRW metric. According to variation $f(R)$ gravity concerning metric, one can have the following equation of motions,

$$3FH^2 = \frac{FR - f}{2} - 3H\dot{F}, \tag{8}$$

and

$$-2F\dot{H} = \ddot{F} - H\dot{F}, \tag{9}$$

where $F = \frac{\partial f}{\partial R}$, and a dot denotes a derivation with respect to $t$. The dynamics of $f(R)$ gravity inflation with the four inflation indicators $\epsilon_i, i = 1...4$ expressed as follows [73–78],

$$\epsilon_1 = -\frac{\dot{H}}{H^2}, \quad \epsilon_2 = 0, \quad \epsilon_3 = \frac{\dot{F}}{2HF}, \quad \epsilon_4 = \frac{\dot{E}}{2HE}, \tag{10}$$

where $E = \frac{3(\dot{F})^2}{2\kappa^2}$. It is necessary to calculate the $Q_s$ to compute the tensor-to-scalar ratio $r$, which is also expressed as the following:

$$Q_s = \frac{E}{FH^2(1 + \epsilon_3)^2}. \tag{11}$$

The spectral index of curvature perturbations $n_s$, is as follows [74–76],

$$n_s = 4 - 2\sqrt{\frac{1}{4} + \frac{(1 + \epsilon_1 - \epsilon_3 + \epsilon_4)(2 - \epsilon_3 + \epsilon_4)}{(1 - \epsilon_1)^2}}, \tag{12}$$

where $\dot{\epsilon}_i \simeq 0$ is assumed. The tensor-to-scalar ratio in the content of modified $f(R)$ gravity theory is expressed as follows,

$$r = \frac{8\kappa^2 Q_s}{F}, \tag{13}$$

the tensor-to-scalar ratio is rewritten as below,

$$r = \frac{48\epsilon_3^2}{(1+\epsilon_3)^2}, \tag{14}$$

where we used Equation (11), and Equation (13).

We can obtain the new relations for the $\epsilon_i$ with respect to Equations (6) and (10) [66],

$$\epsilon_1 = -\frac{\dot{H}}{H^2}, \quad \epsilon_2 = 0, \quad \epsilon_3 = \frac{\dot{F}^{(1)}}{2HF}(24H\dot{H} + \dddot{H}), \quad \epsilon_4 = \frac{F^{(2)}}{HF}\dot{R} + \frac{\ddot{R}}{H\dot{R}}, \tag{15}$$

where $F^{(1)}$ and $F^{(2)}$ are $\frac{\partial^2 f}{\partial R^2}$ and $\frac{\partial^3 f}{\partial R^3}$, respectively. It can be seen that inflationary dynamics are related to $f(R)$ gravity. We are studying the modified $f(R)$ gravity; as we will show in the next section, this model has successfully described the late acceleration period. Additionally, the primordial power spectrum of the curvature perturbation $\zeta$ is described from,

$$\langle \hat{\zeta}(\mathbf{k})\hat{\zeta}(\mathbf{k}') \rangle = (2\pi)^3 P_\zeta(k)\delta^3(\mathbf{k} - \mathbf{k}'),$$

where $\hat{\zeta}(\mathbf{k})$ is the three-dimensional Fourier transform of $\zeta(\mathbf{x})$. The spectral index of this power spectrum $n_s$ is defined as

$$n_s \equiv 1 + \left.\frac{d\ln\left(k^3 P_\zeta(k)\right)}{d\ln k}\right|_{k_*},$$

where $k_*$ is a pivot scale in the observable range, $k_* = 0.05 Mpc^{-1}$ for Planck, a power spectrum increasing on large angular scales ($n_s < 1$) is called red-tilted, if it rises with small scales, it is called blue-tilted, deviation from a scale invariant primordial power spectrum have been detected by recent CMB experiments, the power spectrum is observed to be red-tilted, and the case $n_s = 1$ is today ruled out. The $1 - \sigma$ bound on the spectral index measured by WMAP was [34,79–82] were $n_s = 0.968 \pm 0.012$. Planck has improved by roughly a factor of two in the measurement of the spectral index [2], resulting in $n_s = 0.9649 \pm 0.0042$. On the other hand, the power spectrum amplitude measured by Planck is

$$\mathcal{A}_s \equiv \mathcal{P}_\zeta(k_*) \equiv \frac{k_*^3}{2\pi^2}P_\zeta(k_*) = 2.196^{+0.051}_{-0.06} \times 10^{-9}.$$

Therefore, this structure can be used for deeper investigations, since the amplitude of scalar perturbations $\mathcal{A}_s$ is an observation parameter and can apply additional restrictions on the model parameters. The motivation for slow-roll inflation is that it produces a nearly scale-invariant spectrum of perturbations, compatible with the CMB observations [2,3],

$$\mathcal{A}_s = 2.1 \times 10^{-9}, \quad n_s = 0.9649 \pm 0.0042 \quad \alpha_s = -0.0045 \pm 0.0067, \quad r < 0.036,$$

where $\alpha_s$ running of the scalar spectral index, and $r$ is the tensor-to-scalar ratio at the CMB pivot scale $k_*$. In the slow-roll limit, the perturbations can be computed using the standard formalism, giving

$$\mathcal{A}_s = \frac{V}{24\pi^2\epsilon_V} = \frac{H^2}{8\pi^2\epsilon_H}.$$

By using the given power spectrum, the scalar spectral index can be obtained as follows,

$$n_s = 1 - 2\epsilon - \frac{1}{H}\frac{d}{dt}\ln\epsilon = 1 - 6\epsilon_V + 2\eta_V = 1 - 4\epsilon_H + 2\eta_H, \quad r = 16\epsilon_V = 16\epsilon_H,$$

where we also gave the forms based on the Hubble slow-roll parameters,

$$\epsilon_H \equiv \frac{\dot{\phi}^2}{2H^2} \simeq \epsilon_V, \qquad \eta_H \equiv -\frac{\ddot{\phi}}{H\dot{\phi}} \simeq \eta_V - \epsilon_V,$$

where the approximations apply during slow-roll. The expression for $\alpha_s$ depends on higher-order slow-roll parameters, which we omit for brevity. According to the presented concepts, this time, by using the mentioned limitations, two of the most important parameters of cosmology can be obtained according to the values and free parameters mentioned in the text. Therefore, the scalar spectrum index and tensor-to-scalar ratio, which is obtained as $(n_s = 0.9651, r = 0.00125)$, are compatible with the latest observable data [2,3]. Furthermore, the power spectrum is evaluated at the horizon crossing time, $C_s k = aH$ (with $k$ being the comoving wave number) and given by,

$$\mathcal{A}_s = \frac{H^2}{8\pi^2 \mathcal{W}_s C_s^3},$$

where, by definition, $\mathcal{W}_s \equiv \frac{\dot{\phi}^2}{2H^2}$, and $C_s^2 = 1$. Additionally, the tensor-to-scalar ratio, which takes the following form,

$$r = \frac{\mathcal{A}_t}{\mathcal{A}_s} = 16\epsilon.$$

Up to this point, we obtained the main equations of this inflationary setup. In what follows, performing analysis of their behavior, we obtain some constraints on these parameters through confrontation with the observational viable values of some parameters.

### 3. Logarithmic $f(R)$ Gravity

The conditions and restrictions applied to gravitational models always lead to changes in the features of these models. For example, the constant-roll conditions change the durability of the $f(R)$ gravitational model [66]. The aim of this section is to provide a closer look at our inflationary model as a modified $f(R)$ gravitational model with polynomial plus logarithmic terms and see what happens for the corresponding model. As you know, this kind of $f(R)$ inflationary model has the following form [29,46,70,71],

$$f(R) = R + \alpha R^2 + \theta R^n + \gamma R^2 \ln \gamma R. \tag{16}$$

Here, we note that the logarithmic $f(R)$ gravitational model describes neutron stars, cosmological models, and gluon effects [29–34], where $n$, $\alpha$, $\theta$, and $\gamma$ are constant parameters. The appropriate values of these components lead to solving dimensional problems of the model. It would be reasonable to assume that inflation was predominant in the early universe; thus, we could ignore the contributions of matter and its radiation. According to the above relation, we have,

$$f'(R) = 1 + (2\alpha + \gamma)R + 2\gamma R \ln \gamma R + n\theta R^{n-1}, \qquad f''(R) = 2\alpha + \gamma + 2\gamma \ln \gamma R + n(n-1)\theta R^{n-2}. \tag{17}$$

We can explain the importance of this model according to the above two equations. In the mentioned model, the parameters $\alpha \neq \theta \neq 0$ and only the parameter $\gamma$ is zero, the model examined in Ref. [30]. With $\theta = 0$, the model is reduced to the famous Starobinsky model. A special investigation was also performed with $\gamma = 0$ and $n = 4$ in Ref. [69]. The $f(R)$ also satisfies the conditions $f(0) = 0$ that lead to a flat space-time without a cosmological constant. Additionally, the stability of this model is thoroughly investigated by Ref. [70]. Another critical point is the review of this model in connection with investigating the logarithmic form of the $f(R)$ gravity model from a brane perspective and swampland criteria, which has exciting results that you can see in more detail in Ref. [71]. This model has also been studied in examining a specific type of traversable wormholes concerning various shape and redshift functions. Its exciting results were also investigated in Ref. [72]. In addition, its inflation model has been challenged with the special conditions, i.e., slow-

roll and weak gravity conjecture; the details of this study can be seen in [46]. According to the above model, quantum stability conditions are obeyed by $f(R)$, and also according to the above equations, classical stability conditions also lead to,

$$f'(R) = 1 + (2\alpha + \gamma + 2\gamma \ln \gamma R)R + n\theta R^{n-1} > 0.$$

Now, we consider Equation (8), which for the above inflationary model, it can be approximated as follows,

$$
\begin{aligned}
&-3H^2\left[1 + 18(2+\beta)\alpha H^2 + \theta 6^{n-1}\big((2+\beta)H^2\big)^{n-1}n + 12(2+\beta)H^2\gamma \ln\big(6(2+\beta)H^2\big)\gamma\right] \\
&+\frac{1}{2}\left[-6(2+\beta)\alpha H^2 - 36(2+\beta)^2 H^4 - \theta 6^n\big((2+\beta)H^2\big)^n - 36(2+\beta)^2 H^4\gamma \ln\big(6(2+\beta)H^2\big)\gamma\right] \\
&+6(2+\beta)\alpha H^2\left[1 + 18(2+\beta)H^2\theta 6^{n-1}\big((2+\beta)H^2\big)^{n-1}n + 12(2+\beta)H^2\gamma \ln\big(6(2+\beta)H^2\big)\gamma\right] \\
&-3H\left[1 + 36(2+\beta)\alpha H\dot{H} + 12^{n-1}n\theta\big((2+\beta)H\dot{H}\big)^{n-1} + 24(2+\beta)H\dot{H}\gamma \ln\big(12(2+\beta)H\dot{H}\big)\gamma\right] = 0.
\end{aligned}
\tag{18}
$$

We have used the constant-roll conditions in the above calculations, Equation (6). By performing a series of manipulations and straightforward computations, and with a series of simplifications, we obtain the final relation by solving the differential equation for the Hubble parameter $H$. The general form is as follows:

$$H(t) = \frac{A}{B+C}, \tag{19}$$

where

$$
\begin{aligned}
C &= \exp\left(\frac{\gamma nt\left[24 + 12\alpha\beta + \theta 12^n(2+\beta)^n n\right]\left[1 + 144(2+\beta)^2\big(2^{2n+1}\theta 3^n(2+\beta)^n + 3n\big)c_1\right]}{144(2+\alpha\beta)^2\gamma\big(2^{2n+1}3^n(2+\beta)^n\alpha + 3n\big)}\right), \\
A &= n\left[24 + 12\alpha\beta + \theta 12^n(2+\beta)^n n\right]\gamma, \\
B &= 24n + 2\left[6\alpha\beta n + \theta 6^n(2+\beta)^n\gamma t\big(-36\alpha\beta(2+\beta)^2 + (2+\beta)n - (1+\beta)n^2\big)\right].
\end{aligned}
\tag{20}
$$

Here, we note that $c_1$ is an arbitrary integration constant not affected by inflation dynamics. Using the above equation, i.e., the Hubble rate, we can quickly obtain the slow-roll indices, $\epsilon_i, i = 1...4$. Then, using Equation (15), the indices of slow-roll for our inflation model will be the following,

$$\epsilon_1 = \frac{\dot{H}}{H^2}, \tag{21}$$

and

$$\epsilon_2 = 0, \tag{22}$$

while

$$\epsilon_3 = \frac{\mathcal{A}}{\mathcal{B}}, \tag{23}$$

and

$$\epsilon_4 = \mathcal{C} + \mathcal{D}, \tag{24}$$

where

$$
\begin{aligned}
\mathcal{A} &= (2\beta H\dot{H} + 24H\dot{H})\alpha + \left[5 + 12^{n-2}\theta(n-1)n\big((2+\beta)H\dot{H}\big)^{n-2} + 2\gamma\ln\big(12(2+\beta)H\dot{H}\big)\gamma\right], \\[4pt]
\mathcal{B} &= 2H\big(1 + 18\alpha(2+\beta)\big)H^2 + \theta 6^{n-1}\big((2+\beta)H^2\big)^{n-1}n + 12(2+\beta)H^2\gamma\ln\big(6(2+\beta)H^2\gamma\big), \\[4pt]
\mathcal{C} &= \frac{2\beta H^2\dot{H} + \dot{H}^2}{H\dot{H}}, \\[8pt]
\mathcal{D} &= \frac{12(2+\beta)\left[\frac{\alpha}{3(2+\beta)H^2} + \theta 6^{n-3}\big((2+\beta)H^2\big)^{n-3}(n-2)(n-1)n\right]\dot{H}}{1 + 18\alpha(2+\beta)H^2 + \theta 6^{n-1}\big((2+\beta)H^2\big)^{n-1}n + 12(2+\beta)H^2\gamma\ln(6(2+\beta)H^2\gamma)}.
\end{aligned}
\tag{25}
$$

Furthermore, using the values obtained for slow-roll indices, $\epsilon_i, i = 1...4$ and according to the Hubble rate given in Equation (19) and with straightforward calculations as well as simplifications, the values of these indices can be obtained. Additionally, by using the values of slow-roll indices and even the Hubble rate, we obtain the scalar spectrum index (12) and the tensor to the scalar ratio (14). By performing some manipulations and then calculations, the variation rate of these variables concerning $n$ and $\beta$ are shown in the figures. We also describe these results. As you can see in Figure 1, we plot the rate of change of the scalar-spectrum-index ($n_s$) in terms of the $\beta$, concerning the different values for the component of ($n$) and concerning the constant values of the parameters such as ($\alpha$), ($\theta$), and ($\gamma$). These free parameters ($\alpha, \beta, \gamma$) associated with the model have already been calculated by Ref. [46]. Additionally, they obtained the slow-roll parameters and the most important cosmological parameters, such as the scalar spectrum index and tensor-to-scalar ratio, in Einstein's frame according to the slow-roll conditions. The cosmological parameters were a coefficient of $n$, and applying the latest observable data challenged the changes of these important cosmological parameters according to the swampland conjectures. It was found that the logarithmic model is aligned with the latest observable data. Here too, we studied the compatibility of the scalar spectrum index in terms of parameter $\beta$ and for different values $n$. The compatibility points are specified for specific ranges in the figures. Of course, the changes of these parameters in terms of parameter $n$ and with respect to $\beta$ are also plotted, which shows an acceptable range for this cosmological parameter. This consistency was determined for different values of the parameter $n$ in [46] for the Einstein frame. Of course, the acceptable range for other cosmological parameters, i.e., the tensor-to-scalar ratio $r$ for various values of $n$ in [46] and also for the desired model in Jordan's frame in the upcoming work, is also discussed in separate figures. It is possible to compare the compatibility of these two important cosmological parameters according to two different frameworks. In general, in both frames, a specific range is specified for the constant parameters that are consistent with the observable data. Additionally, the changes in this index in terms of ($n$) are well defined according to the parameter ($\beta$) in Figure 2. As shown in Figure 2b, for ($\beta = -1$), the correct range of this index is displayed, which can be compared with the observable data.

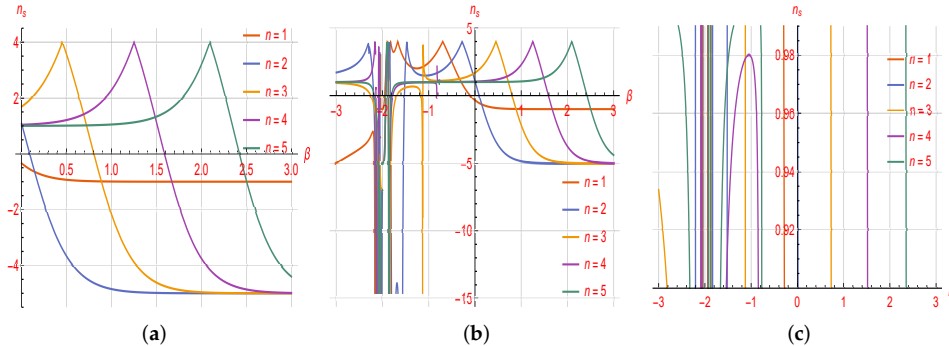

**Figure 1.** The plot of the variation of $n_s$ in terms of $0 < \beta < 3$ in the plot (**a**), $-3 < \beta < 3$ in the plots (**b**,**c**) with respect to different values of $n$ and the constant parameter $\alpha = 0.15$, $\theta = 0.009$, and $\gamma = 0.01$.

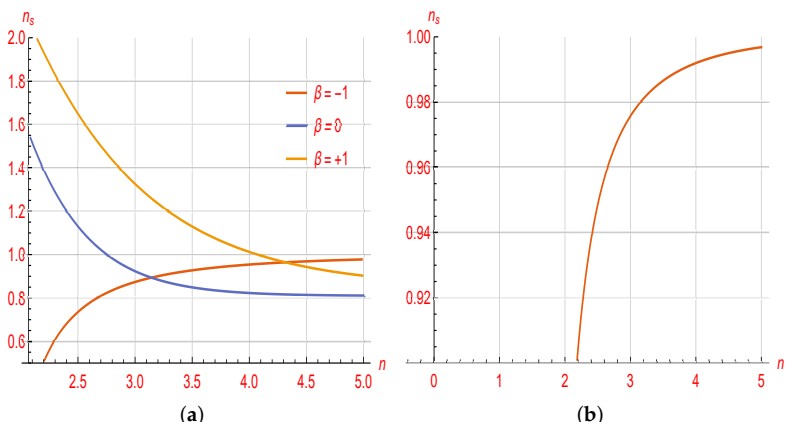

**Figure 2.** The plot of the variation of $n_s$ in terms of n and $-1 < \beta < 1$ in the plot (**a**) and $\beta = -1$ in the plot (**b**) with respect to constant parameter $\alpha = 0.15$, $\theta = 0.009$, and $\gamma = 0.01$.

To understand the physical phenomena in the corresponding model, we took advantage of Equations (12), (18), and (21)–(24), and we plotted the different values of scalar spectrum index $n_s$ concerning various parameters such as $\beta$ and $n$. Here, we note that the variation rate in the above figures is comparable to the experimental data, especially Planck 2018 [2]. Additionally, with respect to the above statement, as can be seen in Figure 3, we also plot the rate of change of the tensor-to-scalar ratio ($r$) in terms of the $\beta$ concerning the different values for the component of ($n$) and with respect to the constant values of the parameters such as ($\alpha$), ($\theta$) and ($\gamma$). The allowable range for this parameter is displayed in Figure 3a for different values of $n$ and it can be seen that it is consistent with Planck's observable data. Of course, the importance of these constant parameters is plotted in each case by keeping the other constant parameter. Furthermore, as you can see in Figure 4, we plot the rate of change of these two cosmological parameters, i.e., the scalar-spectrum-index ($n_s$) and the tensor-to-scalar ratio ($r$) to each other for different values of the parameters ($\beta$) and ($n$) for constant values ($\alpha$), ($\theta$), and ($\gamma$). In Figure 4a,b are well determined the allowable range of these two cosmological parameters ($n_s$) and ($r$) proportional to each of the different values ($n$) and ($\beta$) are well specified. The changes of these two parameters about each other in the Einstein frame were also examined in Ref. [46], which shows the changes of two parameters in terms of constant parameters such as $n$. More precisely, the changes of these two parameters in both forms are very similar, and according to the changes of the free parameters, each of these important cosmological parameters is within the permissible range of the latest observable data. These significant cosmological changes in each diagram can be seen with the changes in free coefficients.

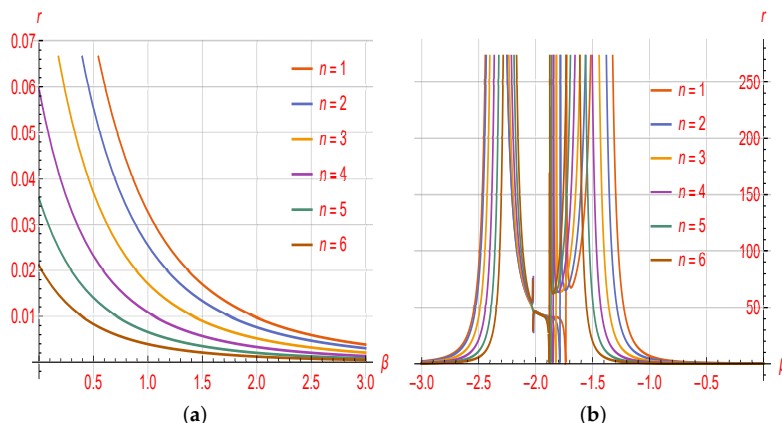

**Figure 3.** The plot of $r$ in terms of $0 < \beta < 3$ in the plot (**a**) and $-3 < \beta < 3$ in the plot (**b**) with respect to various values of $n$ and constant parameter $\alpha = 0.15$, $\theta = 0.009$, and $\gamma = 0.01$.

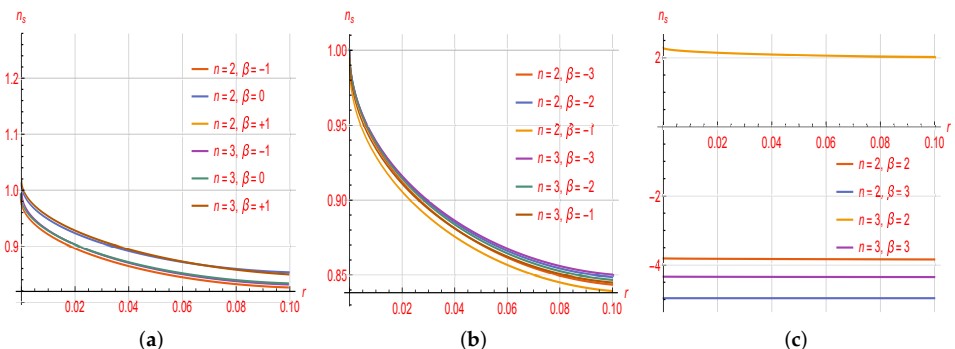

**Figure 4.** The $(n_s - r)$ plan with respect to different values of n and $\beta$ (**a**–**c**) and the constant parameter $\alpha = 0.15$, $\theta = 0.009$, and $\gamma = 0.01$.

Of course, as mentioned above, the modified $f(R)$ gravitational model has been studied under different constraints and conditions, such as the constant-roll and slow-roll conditions. In this article, we take the general form $f(R)$ gravity. It has two exciting terms, polynomial and logarithmic form. It means that we examined the evolution of this model with several parameters. The logarithmic model is one of the essential inflation models that cosmologists have worked on in the last few years. As mentioned above, the scalar spectrum index and the tensor-to-scalar ratio are only based on two constant parameters $\beta$ and $n$. A more detailed analysis shows that their values can be consistent with the observations for many of these parameters. We have given some examples in the form of the above figures.

As a result of cosmic acceleration, small initial velocities within a causally connected patch become very large. In this way, the inflationary paradigm, where the universe undergoes an early-time accelerated expansion, can explain the thermalization of our observable universe and it solves some problems related to the initial conditions of Friedmann cosmology. Inflation can be described by a (quasi) de Sitter expansion where the Ricci scalar is almost a constant and is near the Planck scale. A useful parameter to describe inflation is the e-folds number left to the end of inflation. To examine the manners of the solution during the departure from inflation, we present the e-foldings number as follows [83,84],

$$N = \ln\left[\frac{a(t_f)}{a(t_i)}\right] \equiv \int_{t_i}^{t_f} H(t)dt = \int_{\phi_f}^{\phi_i} \frac{H}{\dot{\phi}} d\phi, \qquad (26)$$

where $a(t_f)$ is the scale factor at the end of inflation with $t_f$—the related time. Therefore, the total amount of inflation is provided by,

$$\mathcal{N} = N\big|_{a(t_i)},\tag{27}$$

where $a(t_i)$ is the scale factor at the beginning of inflation with $t_i$ the corresponding time. To get the thermalization of our universe according to CMB data, one has to require $55 < \mathcal{N} < 65$. During the quasi-dS development of inflation, the Hubble parameter slowly decreases. We defined the slow-roll parameters as follows,

$$\epsilon = -\frac{\dot{H}}{H^2} = \frac{1}{H}\frac{dH}{dN},$$
$$-\eta = \beta = \frac{\ddot{H}}{2H\dot{H}},\tag{28}$$

where we supposed that the constant-roll condition remains valid. At the beginning of the early-time acceleration, the first slow-roll parameter is small. Thus, the slow-roll approximation regime is realized. For the $\beta$ parameter, one can obtain a constant value in the following form [83,84],

$$\beta = \frac{1}{2\mathcal{N}}.\tag{29}$$

This means that the model meets the condition for constant-roll inflation. This issue has an important impact on the form of the spectral index of primordial curvature perturbations, which will be independent of the total number of the e-foldings during inflation. The inflationary paradigm suggests two important forecasts about the inhomogeneities of the universe at a galactic scale. Specifically, the perturbations around the FRW metric conduct a non-flat spectral index $n_s$ and also lead to a non-zero scalar-to-tensor ratio $r$. In the case of $f(R)$-gravity, the inflationary indices have the following form,

$$(1 - n_s) \simeq \frac{2\dot{\epsilon}}{H\epsilon} = -\frac{2}{\epsilon}\frac{d\epsilon}{dN},$$
$$r \simeq 48\epsilon^2.\tag{30}$$

An inflation model must also contain a valid mechanism to depart from the accelerated phase and to lead to the cosmological perturbations at the origin of the anisotropy of the universe. To calculate such perturbations, one computes the spectral index $n_s$ and the tensor-to-scalar ratio $r$. Their values are well determined by the last Planck data. These indexes have to be derived in different ways depending on the theory under consideration, and in what follows, we will furnish their correct expressions in modified gravity and scalar field theories. We can write equations of motion in both scalar fields and modified gravity theories in the following form [83,84],

$$\frac{3H^2}{\kappa^2} = \rho_{eff},$$
$$-\frac{(3H^2 + 2\dot{H})}{\kappa^2} = p_{eff},\tag{31}$$

where $\kappa^2 \equiv 8\pi G_N$, $G_N^{-1/2} = M_{pl} = 1$. $\rho_{eff}$ and $p_{eff}$ stand for the effective energy density and pressure of the universe (in the case of modified gravity they include gravitational terms) fulfilling a continuity equation [83,84],

$$\dot{\rho}_{eff} + 3H(\rho_{eff} + p_{eff}) = 0.\tag{32}$$

For scalar theories, one can obtain

$$\rho_{eff} = \frac{\dot{\phi}^2}{2} + V(\phi),$$
$$p_{eff} = \frac{\dot{\phi}^2}{2} - V(\phi),\tag{33}$$

while for $f(R)$-modified gravity, we find

$$\rho_{eff} = \frac{1}{2\kappa^2}\left((RF - f) - 6H\dot{F} - 6H^2(F - 1)\right),$$

$$p_{eff} = \frac{1}{2\kappa^2}\left((f - RF) + 4H\dot{F} + 2\ddot{F} + (4\dot{H} + 6H^2)(F - 1)\right). \tag{34}$$

Additionally, we can introduce an effective equation of state (EoS),

$$p_{eff} = \omega_{eff}\rho_{eff}, \tag{35}$$

where $\omega_{eff}$ is an effective EoS parameter. At the beginning of inflation, $\omega_{eff}$ must be close to minus one, but not vanishing to depart from early-time acceleration [83,84]. Furthermore, one may need $-1 < \omega_{eff}$ to bypass $\omega_{eff} = -1$ at some time of inflation, since the pure dS solution can be a final attractor of the system. Acceleration vanishes when $-1/3 \leq \omega_{eff}$, namely, the "strong energy condition" (SEC) is violated. Therefore, a suitable ansatz for the effective EoS parameter of inflation in terms of the e-folds may be

$$1 + \omega_{eff} \simeq \frac{\xi}{(N + 1)^\sigma}, \qquad 0 < \sigma, \xi, \tag{36}$$

where $\xi$ is a number on the order of the unit. The outcome, from (32) and (35), by considering that $d/dt = -H(t)d/dN$, is given by

$$\rho_{eff} \simeq \rho_f(N + 1)^{3\xi}, \qquad \sigma = 1,$$

$$\rho_{eff} \simeq \rho_0 \exp\left(-\frac{3\xi}{(\sigma - 1)(N + 1)^{\sigma-1}}\right), \qquad \sigma \neq 1, \tag{37}$$

where $\rho_{0,f}$ stand integration constants. $\rho_f$ exists the effective energy density at the end of inflation at $N = 0$ with respect to $\sigma = 1$, and $\rho_0$ gives the effective energy density at the beginning of inflation at $1 \ll N$ with respect to $1 < \sigma$. Starting from these results, we can now reconstruct the spectral index and the tensor-to-scalar ratio which realize (36) and the corresponding models in the different representations. Thus, the $n_s$ and the $r$ can be obtained as

$$n_s \simeq 1 - \left(\frac{3\xi + \sigma(\mathcal{N} + 1)^{\sigma-1}}{(\mathcal{N} + 1)^\sigma}\right),$$

$$r \simeq \frac{24\xi}{(\mathcal{N} + 1)^\sigma}, \tag{38}$$

where the slow-roll parameters have been evaluated during inflation at $N = \mathcal{N}$, where $\mathcal{N}$ is the total e-folds number. Thus, with respect to the last Planck data, only the case $\sigma = 2$ with $\mathcal{N} \simeq 60$ guides to viable values for the $n_s$ and the $r$. The case where $\sigma = 1$ exists is also quite interesting, since it corresponds to power-law scalar potential [83,84], but, even though it gives a correct value of the spectral index, the tensor-to-scalar ratio is in general larger than the Planck result. For $\sigma = 2$ the EoS parameter (36) with (37) reads,

$$\omega_{eff} \simeq -1 + \frac{1}{9\xi}\log\left[\frac{\rho_{eff}}{\rho_0}\right]^2. \tag{39}$$

Now, we examine the different values of the Hubble parameter according to the concepts mentioned in the text, so first, we calculate the Hubble parameter in terms of $\phi$, which is expressed in the following form,

$$H(\phi) = \Big\{ + 3\gamma^2 n^2 (24 + 12\alpha\beta + 12^n \theta (2\beta)^n n)^2 \times \big\{ 10368(2 + \alpha\beta)^4 (2^{1+2n} \times 3^n \alpha (2 + \beta)^n + 3n)^2$$
$$+ 24n + 2\big(6\alpha\beta n + 6^n \theta (2 + \beta)^n \gamma (-36\alpha\beta (2 + \beta)^2 n - (1 + \beta)(n)^2)\big)$$
$$\Big/ \exp\Big( \frac{n(24 + 12\alpha\beta + 12^n (2 + \beta)^n \gamma n)(144(2 + \beta)^2 (2^{1+2n} \times 3^n \theta (2 + \beta)^n + 3n)\phi)}{72(2 + \alpha\beta)^2 (2^{1+2n} \times 3^n \alpha (2 + \beta)^n + 3n)} \Big) \big\} \Big\}. \tag{40}$$

We know at $\phi = \phi_e$, if one of the slow-roll parameters is in the order one (that is), inflation will end. Furthermore, here we note that in Equations (31) ($\epsilon > \eta$ with respect to $n$). Therefore, for this case, we have the following equation,

$$\mathcal{X}_1 = 36\gamma \ln\Big(\frac{n}{n+2}\Big),$$
$$\mathcal{X}_2 = -2^{(2-\frac{3n}{2})}\alpha \times 3^{1+\frac{n}{2}}\gamma\Big(\frac{n}{1+n}\Big)^{(\frac{-2}{1+n})}\beta(1+n)\Big(\frac{-n}{1+n}\Big)^{-n},$$
$$\mathcal{X}_3 = \Big(4\sqrt{6}n\theta \ln\Big(\frac{n}{n+2}\Big) + 4\sqrt{6}n^2\beta\gamma \ln\Big(\frac{n}{n+2}\Big)\Big),$$
$$\phi_e = \Big(\frac{A}{B \times C}\Big)^{\frac{1}{-n-3}}. \tag{41}$$

Furthermore, according to Equations (29), (31), and (41), we obtain the number of e-folds $N$, which is given by,

$$N = \frac{-2^{\frac{-1}{2}\alpha + \frac{3n}{2}} \times \gamma 3^{-\frac{1}{2}-\frac{n}{2}}\big(\frac{n}{1+n}\big)^{\frac{2}{1+n}}\big(-\frac{n}{1+n}\big)^n \beta \phi_i^{3+n}}{\theta(3+n)(\gamma n + \gamma n^2)} + \frac{4\beta(1+n)}{3(3+n)}. \tag{42}$$

With respect to the two above equations and according to the number of $50 \leq N \leq 60$, we can ignore the second term or the contribution of $\phi_e$. Thus, we have the following equation,

$$\phi_i = \Bigg(\frac{N\alpha\beta(3+n)(\gamma n + \gamma n^2)}{-2^{\frac{-1}{2}\theta + \frac{3n}{2}}\beta \times \gamma 3^{-\frac{1}{2}-\frac{n}{2}}\big(\frac{n}{1+n}\big)^{\frac{2}{1+n}}\big(-\frac{n}{1+n}\big)^n}\Bigg)^{\frac{1}{3+n}}. \tag{43}$$

According to the above equations, the value of the Hubble parameter can be obtained at the beginning of inflation according to Equations (40) and (43) and at the end of inflation according to Equations (40) and (41), which will be calculated very directly. Therefore, by having the values of free parameters such as $\alpha$, $\beta$, $\theta$, $\gamma$, $n$, and the number of e-folds ($N$), it is possible to obtain values of this parameter. In addition, it is possible to reconstruct Starobinsky gravity models by vanishing the free parameters of the model. In fact, our model is a generalized model of $R^2$-Starobinsky model. Furthermore, according to all the concepts above, our model can be compatible with the inflationary paradigm. Moreover, according to Equations (31) and (43), each of the slow roll parameters can be calculated. In addition, with these values, one can calculate each of the important parameters of cosmology, i.e., the tensor-to-scalar ratio and the scalar spectrum index in Equation (33). Similar to the calculations that we performed before for each of these parameters, and according to the figures, we determined the areas compatible with the latest observable data. Furthermore, as we said, by considering zero free parameters, the mentioned model is reduced to the famous $R^2$-Starobinsky and deformed Starobinsky models. Its results are discussed in [26,29,31,32,32,34,85–87]. Therefore, our logarithmic model is a suitable generalization of these models, which can be used in the study of various cosmological structures, its applications and results can be compared with the latest observable data.

## 4. RSC in Logarithmic Constant-Roll Inflation

We investigate the potential of this inflation model concerning the conjecture of the swampland program from the point of view of constant roll. As we know, the swampland dS conjecture is as follows [22–27,46],

$$|\nabla V| \geq C_1 V, \quad min(\nabla_i \nabla_j V) \leq -C_2 V, \tag{44}$$

where the above equations for the $V > 0$ can be rewritten in terms of the slow-roll parameters as follows,

$$\sqrt{2\epsilon_V} \geq C_1, \quad or \quad \eta_V \leq -C_2, \tag{45}$$

where $C_1$ and $C_2$ are both positive and order of one, i.e., $C_1 = C_2 = \mathcal{O}(1)$. $f(R)$ gravity is used for dark energy studies and cosmological models. As it is known, $f$ is a function of the Ricci scale, in which $f = F + R$ [85,86,88–90] is used in cosmological studies and dark energy. Therefore, in this paper, we investigate this inflation model due to refined swampland conjecture and from the point of view of the constant-roll condition. In this case, we apply the modified $f(R)$ gravity and investigate inflationary theory in light of the above. Recently, several researchers have worked with the simple form of $f(R)$, which you can see in Refs. [6,22,91–99]. In general, the constant-roll condition for all inflation models, such as scalar field coupled to gravity and $f(R)$ gravitational models, has been investigated. We aim to investigate the $f(R)$ gravitational model using a constant-roll condition. There will be a special inflation solution for common equations of motion, and we will briefly describe this route. One can begin by considering the general solution for the constant-roll condition in the context of scalar-tensor theory. For the constant roll, we have an important condition in Equations (7). By using the FRW equations,

$$\frac{3}{\kappa^2} H^2 = \frac{\dot{\phi}^2}{2} + V(\phi), \quad -\frac{1}{\kappa^2}(3H^2 + 2\dot{H}) = \frac{\dot{\phi}^2}{2} - V(\phi), \tag{46}$$

we will calculate,

$$\frac{2}{\kappa^2}\dot{H} = \dot{\phi}^2. \tag{47}$$

In the Einstein frame, the Einstein and the Klein–Gordon equations in the lack of the spatial curvature and other matter take the standard form:

$$
\begin{aligned}
H^2 &= \frac{1}{3}\left(\frac{1}{2}(\frac{d\phi}{dt})^2 + V\right), \\
\frac{dH}{dt} &= -\frac{1}{2}(\frac{d\phi}{dt})^2, \\
\ddot{\phi} + 3H\dot{\phi} + \frac{\partial V}{\partial \phi} &= 0.
\end{aligned}
\tag{48}
$$

From the above equations, we will obtain

$$
\begin{aligned}
\frac{d\phi}{dt} &= -2\frac{dH}{d\phi}, \\
\frac{d^2\phi}{dt^2} &= -2\frac{d^2 H}{d\phi^2}\frac{d\phi}{dt}.
\end{aligned}
$$

The second derivative $\ddot{\phi}$ is negligible compared to the other equations above, which is ignored. More precise approximations, and some more precise solutions that give more accurate answers, have been used extensively in the recent examples, especially when we are faced with the specific inflationary potentials $V(\phi)$ that have several non-analytical features [100,101]. If $\frac{\partial V}{\partial \phi}$ is retained for a long time, the second example refers to an ultra-slow-roll model. In agreement with the above equations, Einstein–Hilbert action is

investigated using a canonical scalar field as used in previous works [12,102,103]. Similarly, in $f(R)$, gravitational models are a natural generalization of the constant-roll condition,

$$\ddot{F} = \beta H \dot{F}, \tag{49}$$

where $\beta = -(3 + \alpha)$. $\alpha$ is a non-zero parameter, and for $\alpha = -3$, the model is reduced to the standard slow-roll. Moving beyond the slow-roll approximation, we can consider an ultra slow-roll regime where the $\ddot{\phi}$ is finite in the Klein–Gordon equation as $\ddot{\phi} = -3H\dot{\phi}$. As we know, this condition is in complete agreement with the previous one used in GR and is no different from the previous one. However, as stated in the preceding equation, its generalization is very significant in many respects, and we only use the constant-roll condition for $f(R)$ gravitational models. Using the constant-roll condition concerning the refined swampland conjecture and applying them, we investigate the coefficient of the swampland conjecture with the logarithmic inflation model; we obtain some parameters, such as the potential, by using the above equations, as well as experimental data and Planck 2018 data. Then, we analyze the result of the inflation model. Different inflation models from several perspectives, such as slow-roll, ultra-slow-roll, constant-roll (using methods such as beta and first-order function), and swampland program, including swampland conjectures, etc., have been studied, and examples of them are mentioned in the above remarks. First, we calculate the potential and then the upper bound $n$. We are comparing our inflation model by plotting some figures. We also check whether this gravitational model is consistent with swampland conjectures. Therefore, first, we calculate the potential by using the Hubble parameter Equations (1), (2), and (19) concerning the $f(R)$ constant-roll condition. Thus, one can obtain the final relation for the potential, which is given by,

$$
\begin{aligned}
V(\phi) = \Bigg\{ &- \big(\gamma^2 \exp\big(\tfrac{n(24 + 12\alpha\beta + 12^n(2+\beta)^n\gamma n)(144(2+\beta)^2(2^{1+2n} \times 3^n\theta(2+\beta)^n + 3n)\phi)}{72(2+\alpha\beta)^2(2^{1+2n} \times 3^n\alpha(2+\beta)^n + 3n)}\big)\big) \\
&\times n^4(24 + 12\alpha\beta + 12^n\theta(2+\beta)^n n)^2(24 + 12\alpha\beta + 12^n(2+\beta)^n\gamma n)^2\big) + 3\gamma^2 n^2(24 + 12\alpha\beta + 12^n\theta(2\beta)^n n)^2 \\
&\times \Big\{ 10368(2+\alpha\beta)^4(2^{1+2n} \times 3^n\alpha(2+\beta)^n + 3n)^2\Big[ + 24n + 2(6\alpha\beta n + 6^n\theta(2+\beta)^n\gamma(-36\alpha\beta(2+\beta)^2 n \\
&- (1+\beta)(n)^2)) + \exp\big(\tfrac{n(24 + 12\alpha\beta + 12^n(2+\beta)^n\gamma n)(144(2+\beta)^2(2^{1+2n} \times 3^n\theta(2+\beta)^n + 3n)\phi)}{72(2+\alpha\beta)^2(2^{1+2n} \times 3^n\alpha(2+\beta)^n + 3n)}\big)\Big]^2 \Big\} \Bigg\} \\
&\Big/ \Big( 10368(2+\alpha\beta)^4(2^{1+2n} \times 3^n\alpha(2+\beta)^n + 3n)^2\Big[ + 24n + 2(6\alpha\beta n + 6^n\theta(2+\beta)^n\gamma(-36\alpha\beta(2+\beta)^2 n \\
&- (1+\beta)(n)^2)) + \exp\big(\tfrac{n(24 + 12\alpha\beta + 12^n(2+\beta)^n\gamma n)(144(2+\beta)^2(2^{1+2n} \times 3^n\theta(2+\beta)^n + 3n)\phi)}{72(2+\alpha\beta)^2(2^{1+2n} \times 3^n\alpha(2+\beta)^n + 3n)}\big)\Big]^4 \Big).
\end{aligned}
\tag{50}
$$

After calculating the potential, using the constant-roll condition, we want to consider two conditions of swampland conjectures. Following Equation (44), we aim to determine whether the potential obtained from the above condition is in agreement with them or no. Therefore, concerning Equation (31), we need the first and second derivatives of potential. We challenge the potential changes and the swampland conjecture by plotting some figures, and we will discuss the compatibility or incompatibility of the mentioned model with the swampland conjecture. As shown in Figure 5, from left to right, the potential changes, the first and the second component of the swampland conjecture as $C_1$ and $C_2$ are plotted according to the scalar field $\phi$ and different values of the constant parameter $\alpha$, $\theta$, and $\gamma$. The changes of each of these quantities regarding the constant-roll condition parameter $\beta$ are shown. In the literature, components $C_1$ and $C_2$ are usually constant and positive, and the unit order is such that the second component $C_2$ has smaller values than $C_1$. As it is clear from Figure 5, the first and second components of the swampland conjectures are in their desired range, and also the change of these two components for the various values of the scalar field $\phi$ and the constant parameter $\beta$ is well known. Furthermore, as shown

in Figure 5, the $C_2$ has a smaller value than the $C_1$, and a kind of optimal compatibility of these different conditions is seen.

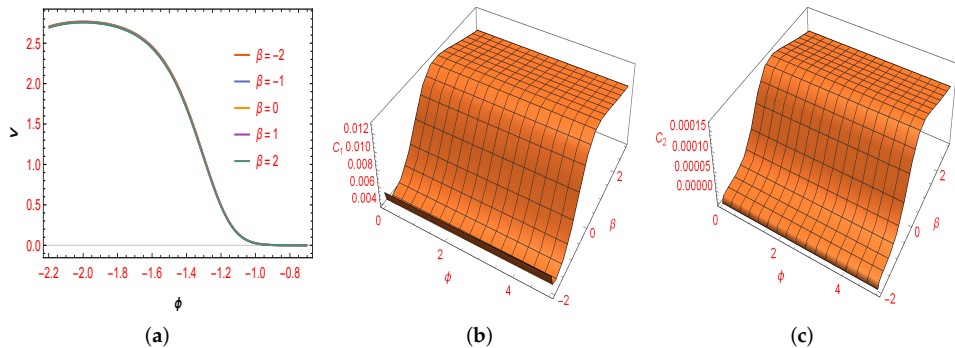

(a)            (b)            (c)

**Figure 5.** The plot of $V$ (**a**), $C_1$ (**b**), and $C_2$ (**c**) in term of $\phi$ with respect to different values of $\beta$, and the constant parameter $\alpha = 0.15$, $\theta = 0.009$, and $\gamma = 0.01$.

Nevertheless, all the above calculations are helpful for these limits to calculations and figures. In this article, we have used one of the conditions for the swampland program. Today, other conditions related to the swampland are more powerful conjectures, such as the trans-Planckian conjecture (TCC), used to investigate the inflation models with certain restrictions. Each of the inflationary models can be examined and studied from the point of view of these conditions.

## 5. Conclusions

A more general and special form of an inflationary model known as the constant-roll condition had been replaced by two-parameter phenomenological inflationary models in *GR* in the slow-roll condition, which was used to study generalized gravitational $f(R)$ models. In this paper, we studied the logarithmic $f(R)$ cosmic evolution with respect to refined swampland conjecture. To review the above, we first introduced our inflationary model, i.e., logarithmic $f(R)$ gravitational model, which is a polynomial function with a logarithmic term. Then, we explained the constant-roll model. We investigated the logarithmic inflation model using constant-roll conditions (constant rate of inflation) and obtained values such as potential and Hubble parameters. We know that the potential value obtained with this condition has an exact value. We examined the constant-roll evolution with logarithmic $f(R)$ gravity. With the constant-roll conditions (18) and performing some manipulations with straightforward calculations and simplifications, we achieved the final relation for the Hubble parameter $H$. This relation helped us to gather more information about the corresponding system. After then, we plotted figures such as $n_s$ with respect to $n$ and $\beta$ separately. Additionally, we plotted the figures $r$ concerning $n$ and $\beta$ and the model's constant parameters, i.e., $\alpha$, $\theta$, and $\gamma$, respectively. In that case, we had some suitable results explained by several figures. Furthermore, we challenged our inflation model with respect to refined swampland conjectures. We concluded that these conditions challenged the swampland conjecture. Finally, we analyzed the figures and evaluated the calculations obtained concerning the experimental data, especially Planck 2018 [2].

**Author Contributions:** Formal analysis, E.N.M.; Writing—original draft, S.N.G.; Writing—review & editing, A.P.; Supervision, J.S.; Project administration, B.P. All authors have read and agreed to the published version of the manuscript.

**Funding:** This research received no external funding.

**Data Availability Statement:** Not applicable.

**Acknowledgments:** We would like to thank Mehdi Shokri for the helpful discussions.

**Conflicts of Interest:** The authors declare no conflict of interest.

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
