# Peer review of "Cosmic Evolution of the Logarithmic f(R) Model and the dS Swampland Conjecture"

_universe, doi:10.3390/universe8120623_

Round 1

Reviewer 1 Report

Referee Report

In the manusript “Cosmic evolution of the logarithmic f(R) model and the dS swampland conjecture», the authors study the inflationary scenarios in logarithmic f(R) gravity, using constant-roll conditions.

Remarks:

1) To calculate inflationary parameters n_s and r the authors do not use the Einstein frame and the potential (32), but works in the Jordan frame and use the slow-roll parameters defined by Eq. (15). The formula for \epsilon_3 in (15) looks strange, because $\epsilon_3=0$ in the case of the $R^2$ Starobinsky inflation and one can get by Eq. (14) that $r=0$. This result is not correct. Maybe there are some typos in formulae and Eq. (23) and the corresponding numerical calculations are not correct. I recommend to calculate the inflationary parameters using the Einstein frame and potential (32) and compare it with the obtained in the Jordan frame.

2) The amplitude of scalar perturbations A_S is an observation parameter and can give additional restrictions on the model parameters. The corresponding formulae should be added.

3) It would better to write out the values of the Hubble parameter at the beginning and at the end of inflation and compare them with the corresponding values of the Hubble parameter in the the case of the $R^2$ Starobinsky inflation.

4) What is the dimension of \gamma? The last summand in formula (16) looks strange. Also, for \gamma R<1, the logarithmic term is negative and can dominate after inflation. The authors should show that the proposed model is close to the Einstein gravity after inflation and does not contradict to observation data.

5) In Fig. 5, it is difficult to see has V(\phi) a maximum or not. The f(R) models described by (16) with $\gamma=0$, have been considered (Q.G. Huang, JCAP 02 (2014) 035 [arXiv:1309.3514]; G. Rodrigues-da-Silva, J. Bezerra-Sobrinho, L.G. Medeiros, Phys. Rev. D 105 (2022) 063504 [arXiv:2110.15502]; V.R. Ivanov, S.V. Ketov, E.O. Pozdeeva, S.Yu. Vernov, JCAP 03 (2022) 058 [arXiv:2111.09058]) and it has been shown for $n>2$ that inflationary scenarios are not realistic, because they require a fine-tuning of the initial value of the scalar field \phi. Does the adding of the logarithmic term solve this problem? Does the scalar field tend to zero during inflation?

Author Response

Many thanks for useful comments of the referee which yield to increasing quality of presentation. Below we give corresponding response point by point.

1- Regarding the first point of the referee's comment, we provide explanations.
The calculations have been done in the Jordan framework because the relevant model has been investigated in different structures and frameworks using other conditions in the  Einstein frame, and the results of it have also been stated in previous works.\\(J. Sadeghi, E. Naghd Mezerji, and S. Noori Gashti, "Study of some cosmological parameters in logarithmic corrected gravitational model with swampland conjectures", Mod. Phys. Lett. A (2020).\\
J. Sadeghi, H. Farahani, "Logarithmic corrected F (R) gravity in the light of Planck 2015", Physics Letters B 751, 89-95 172015 (2015).\\
J. Sadeghi, and S. Noori Gashti, "Investigating the logarithmic form of f (R) gravity model from brane perspective and swampland criteria", Pramana 95 (198) (2021).\\)
Therefore, we compared our calculations, but this time in the framework of Jordan, with other analyses performed by this model in the Einstein frame and different structures and conditions in previous works.
 In the text of the article, we have compared the mentioned issue to some extent with other existing works that are also introduced in the references. Of course, we will add new comparisons by expressing their similarities and differences in red.
Maybe the new calculations in the Einstein frame in this regard are somehow similar to other works, and we want to avoid such overlaps.
Therefore, we will continue the further comparisons by mentioning the details in the Jordan and Einstein frame and other past works by mentioning the references.\\
The dynamics of the f(R) gravity cosmological evolution is determined by the Hubble flow parameters (also known as slow-roll parameters such as $\epsilon_i$). I will introduce some references below to clear the ambiguity and how it is studied in different calculations to check other theories,
Of course, we are also ready to review new recommendations to improve the article's quality.
https://arxiv.org/pdf/1510.04333.pdf\\
https://arxiv.org/pdf/1608.07806.pdf\\
https://arxiv.org/pdf/astro-ph/0107069.pdf\\
S.D. Odintsov et al. / Nuclear Physics B 923 (2017) 608-632

2- Regarding the referee's comment about the scalar perturbations $A_S$, we have added essential concepts with some Refs. before section 3. We also thank the referee for mentioning the valuable point.

3- We express a few points regarding the referee's comment about the  dimension of $\gamma$ and the logarithmic model. ($\alpha$), ($\beta$) and ($\gamma$) are coefficients to solve dimensional problems of the desired model. $\gamma$ is the parameter with the squared length dimension. Also, very comprehensive information on the model has been done in connection with various contents of cosmology, such as inflation in various structures. Regarding the point that the referee also expressed about the sentences of the logarithmic model, one can refer to the following Refs. for further study.\\

J Sadeghi, H Farahani, Logarithmic corrected F (R) gravity in the light of Planck 2015, Physics Letters B 751, 89-95 (2015).\\
J Sadeghi, B Pourhassan, AS Kubeka, M Rostami, Logarithmic corrected polynomial f(R) inflation mimicking a cosmological constant, International Journal of Modern Physics D 25 (07), 1650077 (2016).\\
J Sadeghi, EN Mezerji, SN Gashti, Study of some cosmological parameters in logarithmic corrected  gravitational model with swampland conjectures, Modern Physics Letters A 36 (05), 2150027 (2021).\\
J Sadeghi, M Shokri, SN Gashti, B Pourhassan, P Rudra, Traversable wormhole in logarithmic  gravity by various shape and redshift functions, International Journal of Modern Physics D 31 (03), 2250019 (2022).\\
J Sadeghi, SN Gashti, Investigating the logarithmic form of f(R) gravity model from brane perspective and swampland criteria, Pramana 95 (198) (2021)

4- We will mention a few points according to the referee's comment about Fig. 5. In addition, we added the valuable works introduced by the referee to the references list. Regarding the logarithmic model, the logarithmic sentence gives us valuable points by adding to the polynomial. In the previous works about this model, researchers obtain the constant curvature solutions and found the scalar potential of the gravitational field. They also get the mass squared of a scalaron in Einstein's frame. Other concepts, such as cosmological parameters corresponding to the recent Plank results, are studied. Also, they analyze the critical points and stability of such a modified theory of gravity. Also, they find that both logarithmic and polynomial corrections are necessary to yield slow-roll conditions. And finally, they find that logarithmic correction is required to have a successful model. Additional information about many physical properties of the model, from the potential to the cosmological parameters in different frameworks, is given in the following works.\\
J Sadeghi, H Farahani, Logarithmic corrected F (R) gravity in the light of Planck 2015, Physics Letters B 751, 89-95 (2015).\\
J Sadeghi, B Pourhassan, AS Kubeka, M Rostami, Logarithmic corrected polynomial f(R) inflation mimicking a cosmological constant, International Journal of Modern Physics D 25 (07), 1650077 (2016).\\
J Sadeghi, EN Mezerji, SN Gashti, Study of some cosmological parameters in logarithmic corrected  gravitational model with swampland conjectures, Modern Physics Letters A 36 (05), 2150027 (2021).\\
J Sadeghi, SN Gashti, Investigating the logarithmic form of f(R) gravity model from brane perspective and swampland criteria, Pramana 95 (198) (2021)\\\\

We hope now this revised version be suitable for publication. However, we are ready for any more comments and suggestions.

Reviewer 2 Report

Comments for the paper Universe - 1960465: v1

The paper deals with f(R) gravity, fixing specific types of the function f (and also some conditions on the parameters), so to create the conditions for fitting inflationary scenarios. As it stands, it is a rather standard procedure, for a standard topic.

It is, however, written in a remarkably careless way:
- To begin, the Introduction contains several repetitions of the same concepts
- then, there are incredibly many typos:
1. there is a missing verb in the last sentence of first paragraph
2. "related to the present university" should probably be "related to the present universe"
3. the sentence following to the one in point 3. also has no verb
4. in the sentence "the theoretical framework of f (R), gravity" has a comma that isn't needed
5. in the sentence "ultra-slow-roll conditions [11–41]" there must be a "." at the end of it
6. after eq. 1 we read "are presented as." with a final period which shouldn't be there (this happens many times along the paper)
7. line 75: there must be a capital after a "." (this also happens many times along the work)
8. there are indentations where there should be none (many, many)
9. there should be no capital after commas (also many times)
10. there must be Capitals after full stops (so many)
11. the sentence "above remarks. first We calculate" is a typical example of the fact that punctuation and capital letters are misplaced
12. and so on...
13. and so on...
14. and so on...
- As for the actual science, things are even worse:
1. the Authors indicate the Jordan frame with a j and Einstein frame with an E --- very bad choice: for one thing, j can be confused with an index; then, why does Einstein get a Capital and Jordan a non-Capital?
2. not being enough, the Authors also decide to write derivatives with an index R, adding to the confusion
3. at the beginning they say "the gravitational field equations are presented" but what follows is not the gravitational field equation, just the metric
4. and so on...
At this point I only went on reading without reporting typos and problems: they are just too many, all over, and repeated.
It is even difficult to understand what could have led to this: as for the Introduction I suppose that the several repetitions might come as the results of the fact that each Author might have written one a part and then they simply patched all parts together. For the rest, it looks as if they copy-pasted from a PDF, and the software had a bug at copying.
And in any case, it seems clear to me that none of the Author re-read the manuscript as a final check: no such an amount of errors can pass unnoticed for so many.
In summary, the content of the paper, the methodology and the interest for the community may be of value. But with such an obvious lack of care in the writing, the paper is obviously not acceptable.

Author Response

Many thanks for useful comments of the referee which yield to increasing quality of presentation. Below we give corresponding response point by point.

  • According to the referee's comment about the concepts, typos and grammatical errors, we mention the modified concepts point by point while thanking the reviewer for his precision. First, we read the text several times and checked and corrected similar concepts in various sentences.\\
    1-We have once again thoroughly reviewed the entire article and corrected the content problems and grammatical and spelling mistakes as much as possible with the software. We also modified the last sentence of the first paragraph.\\
    2-This was a big spelling mistake. I thank the referee for his careful opinion. I also corrected this point.\\
    3-We corrected this point.\\
    4-We removed the extra comma.\\
    5-We also fixed the mentioned grammatical problem.\\
    6-By reviewing the entirety of the article, we corrected inappropriate expressions such as (are presented as).\\
    7-We checked and corrected grammar problems such as (.), commas, and lowercase and uppercase letters throughout the article.\\
    8-We modified the existing indentations according to the position of the equations.\\
    9-It is worth mentioning that we fixed such grammatical problems.\\
    10-We corrected the mentioned items.\\
    11-I want to thank the referee for his/her thorough review of the text of the article and the statement of essential problems. We also fixed these mentioned problems.\\
    We completely rewrote the article's introduction, which can increase the quality of the article, and we also thank all the reviewers for their helpful comments. (in red)\\\\
  • This also is one of the spelling problems that we fixed. Of course, to choose the expression and how each of these characteristics is expressed in the equation to introduce different frameworks, we took help from various references.
    https://arxiv.org/pdf/1907.10605.pdf\\
  • In connection with the expression of the derivative concerning an index R, it is a common occurrence in calculations related to effective theories to avoid the large volume of sentences. Of course, before using any structure and command, we also mention its complete explanation, for example, the definition given for these derivatives at the top of equation 10 or the bottom of equation (15).
    In some works, researchers use another methods, such as $(F^{1})$ or $(F^{2})$, which determines the first and second derivatives concerning R. In some works, also is used $'$ or $''$ for the first and second derivatives. We also used numerical indexing to remove the ambiguity and placed it in the text to remove the ambiguity created for the reader.\\\\
  • The referee's point is worthy of consideration since we intended to express the equations completely. Of course, because it did not play a significant role in our calculations, it was not mentioned. We also corrected this sentence. We also thank the referee for his careful opinion.\\\\
  • We rechecked the entire article and corrected the spelling, grammar, and content errors. We again thank the referee for his/her excellent ideas for improving the quality of the paper\\\\

Reviewer 3 Report

The work on the paper is relevant and I recommend it for publication after the author complies with the following minor comments:

The plots in the article should be adequately presented with more prominent readable labels.

Refrain from mentioning variables which are introduced in the text. Please correct some of the grammar in the article.

Author Response

Many thanks for useful comments of the referee which yield to increasing quality of presentation. Below we give corresponding response point by point.

1- According to the referee's opinion, we have changed the format of all plots. Now the characters and labels of the diagrams are much clear. Thanks to the referee for the comment that led to the improvement of the quality of the article.

2- We once again reviewed the entire text of the article and checked and corrected all the grammatical errors and spelling problems. We also removed the variables that did not need to be repeated in the text. This comment of referee was also very appropriate.

Round 2

Reviewer 1 Report

Referee Report

The revised version of the manuscript “Cosmic evolution of the logarithmic f(R) model and the dS swampland conjecture» is not suitable for publication. The authors really do not take into account my remarks. Also, I should note that the very strange Eq. (3) shows that the authors wrote the article very carelessly. The article should be rejected.

Remarks:

1) My remark was: “To calculate inflationary parameters n_s and r the authors use the slow-roll parameters defined by Eq. (15). The formula for \epsilon_3 in (15) looks strange, because $\epsilon_3=0$ in the case of the $R^2$ Starobinsky inflation and one can get by Eq. (14) that $r =0$. This result is not correct.”

In the revised version I see again that \epsilon_3 is proportional to

\dot{F^(2)}=\dot{f^(3)}. It is equal to zero for f=R+\alpha*R^2 (the Starobinsky model), therefore, r=0 in the Starobinsky model (Eq. (14)). It is not correct. The authors did not correct Eq. (15).

2) My remark was: “The amplitude of scalar perturbations A_S is an observation parameter and can give additional restrictions on the model parameters. The corresponding formulae should be added.”

The authors add only the definition of A_S, do not include formulae that demonstrate how A_S depends on parameters of the model considered and the corresponding restrictions on the model parameters. Without these restrictions it is not correct to say that the considered model does not contradict to the observation data.

3) The authors ignored my remark “It would better to write out the values of the Hubble parameter at the beginning and at the end of inflation and compare them with the corresponding values of the Hubble parameter in the case of the R^2 Starobinsky inflation.” I do not see these values of the Hubble parameter (in the Jordan frame) in the revised version of the paper.

4) On my remark was: “What is the dimension of \gamma? The last summand in formula (16) looks strange. Also, for a positive \gamma and \gamma R<1, the logarithmic term is negative and can dominate after inflation. The authors should show that the proposed model is close to the Einstein gravity after inflation and does not contradict to observation data.

The authors reply that $\gamma$ is the parameter with the squared length dimension. I agree with it. But in this case captures of all Figures looks strange as well as Eq. (32). It is difficult to understand to fix the authors the value of the Planck mass or not. Looking at Eqs. (1) and (16) one can assume that M_{Pl}=1, but in Eq. (26) the authors include M_{Pl} without any comments about it value. The authors do not show that after inflation the proposed model is close to the Einstein gravity.

5) The authors ignore my question about the existence of maximum of the potential (32) as a function of \phi.

Author Response

We have revised our paper entitled "Cosmic evolution of the logarithmic $f(R)$ model and the dS swampland conjecture". We have done our best to revise the paper appropriately and respond to referee's comments in the following form:\\\\
\vspace{0.8cm}
\noindent

\textbf{\textcolor{red}{1st referee comment}}:\\

1) My remark was: "To calculate inflationary parameters $n_s$ and r the authors use the slow-roll parameters defined by Eq. (15). The formula for $\epsilon_3$ in (15) looks strange, because $\epsilon_3=0$ in the case of the $R^2$ Starobinsky inflation and one can get by Eq. (14) that $r =0$. This result is not correct."
In the revised version I see again that $\epsilon_3$ is proportional to
$\dot{F^(2)}=\dot{f^(3)}$. It is equal to zero for $f=R+\alpha*R^2$ (the Starobinsky model), therefore, r=0 in the Starobinsky model (Eq. (14)). It is not correct. The authors did not correct Eq. (15).
\\\\

\textbf{\textcolor{blue}{1st authors reply}}:\\

The referee's point is correct; in fact, we modified all the equations in the way that $F=df/dR$, $F^{1}=d^{2}f/dR^{2}$ and $F^{2}=d^{3}f/dR^{3}$. It was a typo that has been resolved.\\\\

\textbf{\textcolor{red}{2nd referee comment}}:\\

2) My remark was: "The amplitude of scalar perturbations $A_S$ is an observation parameter and can give additional restrictions on the model parameters. The corresponding formulae should be added."
The authors add only the definition of $A_S$, do not include formulae that demonstrate how $A_S$ depends on parameters of the model considered and the corresponding restrictions on the model parameters. Without these restrictions it is not correct to say that the considered model does not contradict to the observation data.
\\\\

\textbf{\textcolor{blue}{2nd authors reply}}:\\

Many thanks referee to stress this point. Yes, the study of scalar and vector perturbations can be an important option for the study of important parameters of cosmology, which we also decided to challenge in the separate work of complete perturbations of the desired model. However, by adding the mentioned relations, we obtained two of the most important cosmological parameters and compared them with the latest observable data, where the changes are marked in blue (see the pdf with the name "highlight" where blue lines are after the Fig.4).\\\\

\textbf{\textcolor{red}{3rd referee comment}}:\\

The authors ignored my remark "It would better to write out the values of the Hubble parameter at the beginning and at the end of inflation and compare them with the corresponding values of the Hubble parameter in the case of the $R^2$ Starobinsky inflation." I do not see these values of the Hubble parameter (in the Jordan frame) in the revised version of the paper.\\\\

\textbf{\textcolor{blue}{3rd authors reply}}:\\
Sorry for the confusion. Indeed we thought it is optional as referee told "it would better...", it was the cause of ignoring this point (although we wrote in the comment to editor about this point).
Regarding the referee's comment, we have added now some new calculations and explanations in blue color to the text (see the end of section 3), which we hope that the added content can have as comprehensive and complete explanations as possible for the readers and the reviewer. In that case we have more comparison with the $R^2$ Starobinsky inflation and logarithmic-corrected $R^{2}$ gravity. Of course, the higher accuracy measured is due to the presence of more free parameters in the model, which can be obtained by manually adjusting these parameters to get acceptable values with the latest observable data. Of course, as we mentioned in the text and according to the restrictions applied to the model and the calculations done in the Einstein framework for the model mentioned in Refs., we were able to obtain the bounded logarithmic model free parameters. However, the calculated values are almost aligned with the observed values. Of course, by applying more restrictions, a more accurate value of each of the other important parameters of cosmology can be obtained in connection with the mentioned model, among which the investigation of scalar and vector perturbations can be stated, which will be an attractive topic for further study of the model.\\\\

\textbf{\textcolor{red}{4th referee comment}}:\\

4) On my remark was: "What is the dimension of $\gamma?$ The last summand in formula (16) looks strange. Also, for a positive $\gamma$ and $\gamma R<1$, the logarithmic term is negative and can dominate after inflation. The authors should show that the proposed model is close to the Einstein gravity after inflation and does not contradict to observation data."
The authors reply that $\gamma$ is the parameter with the squared length dimension. I agree with it. But in this case captures of all Figures looks strange as well as Eq. (32). It is difficult to understand to fix the authors the value of the Planck mass or not. Looking at Eqs. (1) and (16) one can assume that $M_{Pl}=1$, but in Eq. (26) the authors include $M_{Pl}$ without any comments about it value. The authors do not show that after inflation the proposed model is close to the Einstein gravity.
\\\\

\textbf{\textcolor{blue}{4th authors reply}}:\\

We correct the problem of $M_{Pl}$ value and unify it whole the paper, and give more explanation about comparison of our model with previous famous one (see comments after (25)). 

\textbf{\textcolor{red}{5st comment}}:\\

5) The authors ignore my question about the existence of maximum of the potential (32) as a function of $\phi$.
\\\\

\textbf{\textcolor{blue}{5st  reply}}:\\

We plotted the potential diagram in $2D$ form according to the scalar field and other constant parameters, as seen in Fig. 5.\\\\

We hope we have been able to  properly address the referee's useful
comments, so that this revised version be suitable for publication.\\\\
Best regards\\
Authors

Reviewer 2 Report

I have seen that the authors have somehow corrected the paper, although there are still typos (such as, for example, a sentence finishing with "as follows." and after the "." there is an equation --- there are a few others like these, but I think it is no longer my role to point them out one-by-one): I must admit that if this paper were written by one of my co-authors I would feel a little uneasy to post it on-line, but I guess that in today's world it is fine. So I will suggest acceptation. However, I really encourage all of you authors to check it again, and/or ask the typesetters to do it for you.

Please understand it is not for me, it is for you... when this paper will be publicly available, and others will see all the typos, it will be your presentation, not my review, that will be judged.

Author Response

Dear respected referee,

Again we apologize for the several typos and grammatical errors. It was due to the short deadline for submitting the article for the special issue, so this paper was prepared quickly and most of the time was spent on the calculations. However, we tried to solve language and editorial errors in this revised version.

Best

Authors